# OpenReview forum: "Benchmarking LLM Guardrails in Handling Multilingual Toxicity"
_TMLR — Rejected by TMLR_

### Review · Reviewer_yfiK · 2025-08-05

**Summary Of Contributions:**

The paper introduces a multilingual safety test suite (7 datasets, 10+ languages) and benchmarks four open-source guardrails (Llama-Guard-2/3, Aegis-Defensive, MD-Judge). It finds substantial performance drops on non-English inputs especially low resource langauges and observes brittleness to multilingual/code-switching jailbreaks.

**Audience:**

Yes

**Audience Explanation:**

Yes.

**Important problem**

The paper directly targets the underexplored gap of multilingual safety for LLM guardrails, building a seven-dataset, 10+ language test suite and probing jailbreak/code-switching robustness.

**Breadth of experiments/ablations**

It benchmarks four popular guardrails across multiple datasets and languages, analyzes non-English performance drops, studies prompt tuning effects, resource-level effects, with additional results in the appendix.

**Broader Impact Concerns:**

No major concerns.

**Claims And Evidence:**

Yes

**Claims Explanation:**

The paper has several experiments and ablations showcasing that the LLMs are not robust to multilingual prompts, thus eliciting unsafe responses. However, the evaluation setting and the translation quality are concerning.

**Off-the-shelf benchmarking**

The paper evaluates existing guardrail classifiers (Llama-Guard-2/3, Aegis-Defensive, MD-Judge) rather than evaluating a stronger multilingual safety model. As the existing guardrails are good in English, but might perform worse in other langauges, it is not clear whether the drop in performance and false positives are due to low inherent multilingual performance of the model or safety.

**Translation and verification**

Large portions of the suite come from machine-translating English datasets via Google Translate, and the authors explicitly note this reliance as a limitation without human validation. It will be good to run some human validation on the susbet of test-suite to validate its quality.

**Language coverage**

Coverage is limited and uneven (e.g., only a subset of ToxicChat translated; a small split for PTP). Broader inclusion especially of lower-resource languages would strengthen the work.

**Requested Changes:**

See above.

---

### Review · Reviewer_V6Ly · 2025-08-05

**Summary Of Contributions:**

The paper introduces a multilingual test suite to benchmark the performance of SOTA guardrail models and analyzes the performance of these models on toxic prompts in multiple languages. The paper shows that the performance of the guardrails on multilingual prompts is significantly below their English performance.

Strengths and weaknesses:
Strength: the paper tackles a very important question in LLM safety research — the robustness of the guardrails on multilingual datasets.

Weaknesses: The paper leaves open alternative explanations for the observed findings and generally, for an analysis paper, the analyses are lacking depth.

**Audience:**

Yes

**Audience Explanation:**

The paper benchmarks the performance of SOTA guardrail models on a multilingual dataset and shows that the performance is below that on the English benchmarks. I think the finding would be of interest to practitioners in safety research and multilingual work.

**Claims And Evidence:**

No

**Claims Explanation:**

The paper claims that the performance of the guardrails on multilingual prompts is significantly below that on English prompts. However, I see two alternative hypotheses for the findings: translation quality of the dataset is poor and the LLM backbones of the guardrail classifiers perform poorly on the languages tested overall (and not specifically on the toxic prompts). Both of these hypotheses, if true, would lead to the observed findings but would require a different approach to improve performance compared to underperforming guardrails and should therefore be investigated. I detail these hypotheses below.

**Requested Changes:**

Critical:
1. Translation quality:
a) The first contribution of the paper is to create a multilingual test bench for guardrail evaluation. To do so, the authors collect publicly available datasets in English and translate them into 8 languages. However, no measures of translation quality are provided. Thus, it’s unclear whether poor performance on these languages can be explained through translation quality. And generally, for a paper the contribution of which is to introduce a benchmark, providing measures of benchmark quality is a must. At a minimum, the authors should report the standard translation quality metrics like BLEU score, etc. Another, more convincing, option would be to translate some of the multilingual benchmarks from English into the target languages following the same protocols as for the monolingual datasets and benchmarking the models both on natural and translated data to show that the performance on the safety datasets cannot be fully explained by the quality of translation.

b) Additionally, I have some smaller questions about the benchmark construction. Some of the listed datasets are already multilingual based on Table 1 and Table 7. How were these datasets handled? Were the 8 languages for the monolingual translations added to these datasets or were the original multilingual datasets added to the benchmark as is? Basically, it’s unclear to me if we’re comparing the same languages across different datasets.

2. Target language choice and multilingual performance: What is the motivation for focusing on the specific languages and models in the study? The reason I’m asking is because out of 8 languages tested, Llama-Guard-3 only explicitly supports English and German. Llama-Guard-2 is an earlier version of the model and Aegis is based on yet an earlier version of Llama-Guard and thus it’s reasonable to assume that these models either support the same languages or fewer languages. If a model explicitly doesn’t support a particular language, wouldn’t one trivially expect poor performance? Wouldn’t a better test be to benchmark the model on the languages it supports?

More generally, all of the safety classifiers tested in the paper rely on an LLM backbone and thus the performance of the classifier on multilingual prompts will be affected by the LLM’s performance on a particular language (if the LLM does poorly on this language, there’s little reason to expect that a safety classifier on top of the LLM would do well). And it’s difficult to make claims about the guardrails’ multilingual performance without knowing the backbone’s performance on the specific languages tested or without having explicit information on the languages supported by the guardrail models. I think some benchmarking of the LLM backbones on standard NLP tasks is necessary for the languages of interest to rule out a possibility that the drop in performance on non-English prompts is attributable to the backbone’s overall performance on languages other than English rather than being specific to the guardrails.

3. Analyses: I find it difficult to understand the analyses for a variety of reasons:
	a) Table 1 lists 7 datasets as a comprehensive benchmark but no analysis is conducted on all of them and as far as I can tell PTP is never used. Why?
	b) In all tables, what is the Mul column — an average of all non-English languages? Since some of the datasets are multilingual already, are we even comparing apples to apples in Table 1 (and elsewhere) or are we looking at the averages of different languages, some of which were translated with Google translate and others obtained in a different manner?
	c) More generally, for all datasets I’d like to see a split by language to understand the degree of variability. I think this would also be useful for a practitioner trying to assess whether a given model could be employed for a particular language. Currently, it’s very difficult to tell since the by-language splits are provided only for some datasets. I’d suggest to provide some aggregate statistics including some measure of variability like standard error of the mean in the tables and including the individual language performance into the appendix.
	d) Is Aegis and AegisSafety the same dataset?
	e) I don’t understand the difference between RQ1 and RQ2 on p3. Doesn’t the fact that the prompts are coming from different sources also means various prompts?
	f) For the code-switching analysis, what are the proportions of non-English inserted? Are they the same across all languages? Here, I’d like to see this analysis as well as the split by language.
	g) The analysis of low-med-high resource languages is unconvincing. First, it’s only done on one model (Aegis which unsurprisingly does better on the Aegis dataset) rather than 4; on a subset of datasets and aggregated over languages with no statistics on the variability.

4. It’s unclear to me how Multijail and AutoDAN are different from the other datasets.
5. I don’t quite understand the discussion of the default and customized policies.

Smaller comments:
Tables 1 and 7 overlap in information. I’d suggest merging them.
The paper is well within the page limit, I’d suggest integrating the appendix in the current form into the main text and reserving the appendix for additional in-depth analyses.
The bar charts have different values on the y-axes which makes it easy to misinterpret model performance at a quick glance. I’d suggest limiting the y-axis to (0, 100) throughout.
I find Fig. 1 uninformative since it’s illustrating a concept that can be straightforwardly conveyed in 1 sentence. I’d remove it and use the space to elaborate on some of the points I raised above.

---

### Review · Reviewer_xL7F · 2025-08-24

**Summary Of Contributions:**

This paper tackles an important gap in LLM safety research by systematically assessing how effectively existing guardrails manage toxic content in multiple languages. The authors develop a comprehensive test suite covering seven datasets and more than ten languages, then benchmark four state-of-the-art guardrails on their multilingual performance.

The main contributions include:
- creating the first systematic multilingual evaluation framework for LLM guardrails,
- showing significant performance drops of current guardrails on non-English content,
- exposing vulnerabilities to multilingual jailbreaking attacks, especially code-switching techniques, and
- pinpointing key factors that affect multilingual safety, such as language resource availability and in-context safety policies.

The work addresses an important and underexplored problem with practical implications for deploying LLMs globally. The evaluation is thorough, covering multiple datasets, languages, and attack scenarios. The systematic analysis of factors influencing performance offers valuable insights for practitioners. The evaluation of code-switching attacks is particularly innovative and uncovers concerning vulnerabilities.

The study is limited to open-source guardrails only, possibly overlooking key commercial solutions. Heavy reliance on Google Translate for creating multilingual datasets raises concerns about translation quality and potential artifacts. The analysis does not include statistical significance testing or a deeper exploration of why performance drops across languages occur.

**Audience:**

Yes

**Audience Explanation:**

This work addresses a critical safety concern that directly impacts the real-world deployment of LLMs in multilingual settings. As LLMs become increasingly global, understanding their safety limitations across languages is essential for both researchers and practitioners. The findings reveal significant gaps in current safety measures that could have serious implications for users speaking non-English languages.

**Broader Impact Concerns:**

The paper addresses important safety concerns and includes an appropriate ethical statement acknowledging potential risks.

**Claims And Evidence:**

Yes

**Claims Explanation:**

The paper presents consistent evidence across multiple datasets and models showing that guardrails perform worse on multilingual content. The trends are clear and reproducible across different experimental settings. The F1 scores demonstrate substantial performance drops, and the vulnerability to code-switching attacks is convincingly demonstrated.

**Requested Changes:**

Provide more detail (or a supplement) on how the multilingual datasets were assembled, especially the translation process. For example, were translations manually checked? Adding a few examples or error analysis of translated toxic sentences would strengthen confidence in the evaluation. Additionally, consider evaluating translation quality using modern multilingual LLMs as either an alternative to Google Translate or as a validation method. A simple comparison study showing translation quality differences between Google Translate and state-of-the-art multilingual LLMs on a subset of the toxic content would help establish whether the observed performance drops are due to genuine multilingual safety gaps versus translation artifacts. This could involve back-translation validation, human evaluation of a small sample, or automated quality metrics comparing different translation approaches.

---

### Decision · Action_Editor_c61R · 2025-10-17

**Recommendation:** Reject

**Audience:**

Yes

**Audience Explanation:**

Yes, the topic of multilingual LLM safety is relevant to TMLR’s audience

**Claims And Evidence:**

No

**Claims Explanation:**

This paper benchmarks LLM guardrails for multilingual toxicity detection. While the topic is timely and important for safety in multilingual LLMs, the reviewers raised major concerns about: (1) the lack of deep exploration and analysis of the empirical results; (2) the heavy reliance on Google Translate for creating the multilingual dataset, raising questions about translation quality.

The authors did not submit a rebuttal, and thus left these major concerns unresolved.